# Effect of Thermal Treatment on Nano- and Micro-Copper Particles for Jewelry Making

**Pimthong Thongnopkun** [1,*], **Aumaparn Phlayrahan** [2], **Dawan Madlee** [2], **Worachai Roubroumlert** [2] **and Matinee Jamkratoke** [3]

1 Faculty of Science, Burapha University, Chonburi 20131, Thailand
2 Faculty of Gems, Chanthaburi Campus, Burapha University, Chanthaburi 22170, Thailand
3 Faculty of Science and Arts, Chanthaburi Campus, Burapha University, Chanthaburi 22170, Thailand
* Correspondence: pimthong@go.buu.ac.th

**Featured Application: Nano- and micro-copper particles can be used in artworks, handmade jewelry, ceramic slip, conductive ink, and electronic devices.**

**Abstract:** Copper nanoparticles are being applied in the biosensors, engineering, electronic devices, and technology fields. A key advantage of nanomaterials is that their properties differ from their bulk with the same composition. However, the application of nano-copper particles for jewelry and artwork has not yet been revealed. We discovered and compared the application of nano-sized copper particles in jewelry fabrication with micro-copper particles. The nano- and micro-sized copper particles were synthesized and mixed with the same organic binders and water content to produce an alternative clay-like material for creating handmade jewelry and artwork. The article addresses the effects of thermal treatment on thermal behavior and the development of physical properties of differently sized copper particles. Their physical properties depend strongly on the size of the starting copper particles and heating conditions. We investigated the influence of thermal treatment, heating rate, firing temperature, and holding time to optimize the firing conditions for jewelry fabrication and wearing.

**Keywords:** thermal treatment; nano-copper particle; copper clay; metal clay; jewelry

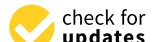



## 1. Introduction

In jewelry manufacturing, there are several common processes for producing jewelry, such as the die-struck, machine-made, and casting techniques. These processes require craftsmen, employees, specific machines, and long production processes. Generally, factory-manufactured jewelry involves mass production, producing numerous identical jewelry products. Therefore, the demand for handmade jewelry is growing due to its uniqueness and attractiveness, mores than the machine-made technique. Precious metal clay is an alternative material for creating handmade jewelry and is suitable for creating small decorative objects, small sculptures, or artworks. The clay is a crafting medium of microscopic metal particles, i.e., silver, gold, or copper, mixed with some organic binders and water. It can easily be molded, shaped, modeled, or sculpted using only hands or simple tools because of a clay-like features of the metal clay [1,2]. After drying, the metal clay can be fired either by a kiln, a handheld gas torch, or a gas stove, depending on the type of clay. The moist organic binder can be removed during the firing process and become a bulk sintered metal. Silver was the first metal to produce metal clay launched in Japan in 1990 [2]. Although there are many types of precious metal clays today, only a few manufacturers manufacture them via different formulas. Normally, they are manufactured from micro-sized metal particles with individual mechanical properties and required firing techniques.

Hypothetically, the nano-sized metal particle is considered a novel material because the reduced particle-sized can decrease the melting point of the metal. In addition, the nano-sized material is expected to advocate a dense structure and suitable mechanical properties, particularly at low temperatures [3–5]. During the process, we successfully produced nano-silver clay utilized for jewelry applications [4]. The nano-silver clay has sintering temperatures significantly lower than micro-silver clay, and the obtained nano-silver jewelry exhibited desirable mechanical properties for wearing jewelry. Although silver clay has promising properties for handmade jewelry, it is rarely available on a commercial scale for the jewelry industry and artwork and is expensive. Accordingly, copper clay is an alternative material for creating handmade jewelry, small sculptures, and artworks because it is less expensive than metal clay. Some reports have revealed the promising properties of nano-copper particles in interconnecting applications, biosensors, and electrochemical sensors, and they supper hard coatings for electronic device production [6–8]. However, the application of nano-copper clay for jewelry or artwork has not yet been revealed.

In this article, we propose nano-copper particles for nano-copper clay, a new jewelry fabrication material. Since both nano- and micro-sized metal particles can be utilized for creating metal clay [4], we assessed nano- and micro-copper clays' fabrication capability for jewelry applications. It is well known that heating parameters dramatically affect finished metal clay jewelry's physical and mechanical properties. Our objective for this research was to determine the effects of thermal treatment on some physical and mechanical properties of both nano- and micro-sized copper particles in clay material. Since wearable jewelry fabricated from metal clay is committed to various physical properties, i.e., hardness, shrinkage, and density of the fired specimen, we determined and compared the suitable firing conditions for nano- and micro-copper clays in this study. In addition, firing conditions, such as heating temperature, heating rate, and holding time, were examined to determine the optimal firing conditions. Furthermore, we conducted DTA analysis and captured SEM images of the fired specimens to determine the microstructure of the sintered specimens.

## 2. Materials and Methods

### 2.1. Materials

The copper particles were synthesized by the chemical reduction method in an aqueous solution. Copper sulfate pentahydrate ($CuSO_4 \cdot 5H_2O$, Merck), ascorbic acid ($C_6H_8O_6$), and polyvinylpyrrolidone (PVP, laboratory-grade, of the average molecular weight of 40,000) were used as a precursor, reducing agent, and capping agent, respectively. All chemicals were analytical grade and used as purchased without further purifications. The synthesis of copper nanoparticles began by adding 250 mL of 0.1 M copper sulfate pentahydrate solution to 250 mL of PVP (0.1%) solution with vigorous stirring for 30 min. Then, 200 mL of 0.2 M ascorbic acid solution was added to the mixture under continuous stirring, followed by slowly adding 1.0 M sodium hydroxide to adjust the solution to pH 7. The prepared solution was continuously stirred at a constant rate of 400 rpm with heating at 80 °C for 2 h. The color of the solution turned from blue to brick red and then reddish brown. After completion of the reaction, the solution was removed from the heat source, and the precipitate was separated by filtration and washed with deionized water three times. Under these conditions, average particle size of the copper nanoparticles was controlled in the range 120–200 nm. To synthesize micro-copper particles, copper sulfate pentahydrate solution was distilled in deionized water without the PVP capping agent, with concentrations of the other reagents and the reaction process the same as for the nano-copper synthesis. Under this condition, average size of the copper micro-particles ranged 3–5 microns. After drying, the micro- and nano-particles were stored in glass vials for further copper clay production. The particle sizes were confirmed by a scanning electron microscope (SEM).

## 2.2. Preparation of Nano- and Micro-Copper Clay

The synthesized nano- and micro-sized copper particles were mixed with the same types and weight ratio of organic binders and water to produce nano- and micro-copper clays. Sodium dodecyl sulfate (SDS) and methylcellulose (MC) were used as the clay's organic binder. Nano- and micro-copper clays were prepared using the same weight ratio of copper powder, SDS, MC, and water at 9.6: 0.1: 0.3: 2, respectively, to yield a clay-like material. Regarding our knowledge of the production process of nano-silver clay [4], type and weight percentage of binders in both copper clays were designed to obtain nano- and micro-copper clays. The unfired clay specimens were wrapped in Parafilm to prevent evaporation and oxidation from the atmosphere. Then, the synthesized nano- and micro-copper clays were molded into a square ($10 \times 10 \times 2$ mm) form to obtain the unfired clay specimens of both materials. After being dried at room temperature for 24 h, the heating experiments were conducted. To verify optimal firing conditions and to study the effects of copper particle size, heating conditions were controlled with temperatures ranging from 200 to 1000 °C, with a heating rate of 10 to 60 °C/min and holding time from 0 to 60 min.

## 2.3. Measurements

The thermal profile measurement and weight loss of the nano- and micro-copper clays during the heating process were determined by a simultaneous thermogravimetric analyzer (STA449C Jupiter, NETZSCH, at National Metal and Materials Technology Center Thailand). The clays were analyzed at temperature scans up to 1000 °C and a heating rate of 30 °C/min. The remaining chemical composition and SEM images obtained from the fracture surface of the sintered clays were monitored using JEOL JSM-6510A scanning electron microscopy equipped with an EDX Genesis energy dispersive x-ray elemental analysis system (SEM-EDS). Physical properties of the fired products important for jewelry wearing were investigated, including volume shrinkage, hardness, porosity, and density. The volume shrinkage was measured using the dimensioning method. Vickers microhardness measurement was assessed using the Universal Hardness DIA-Testor Model 751 with 5 g of applied indenter load and 5 s of dwell time. The bulk density measurements were performed based on the Archimedes principle.

## 3. Results and Discussion

### 3.1. Thermal Behavior of Micro- and Nano-Copper Clay

Figure 1 shows the TG–DTA curves of nano- and micro-copper clays during the heating process. As illustrated, with a solid line in Figure 1, the TG curves of both clays revealed two weight loss processes corresponding to the 0.5% loss of water at 90 °C and the 2.5% loss of organic binders at a broad range of 200–450 °C. The total weight loss of TG curves is related to the total weight of added organic binders and water in the clays. The DTA curves of nano-and micro-copper clay during the heating process were shown as a dashed line in Figure 1a,b. Both curves indicated two endothermic peaks at 90 and 420 °C, corresponding to water evaporation and the decomposition of organic binders, respectively. The DTA analysis of nano-copper (Figure 1a) revealed an exothermic peak related to the sintering process of nano-copper clay occurring at a low temperature, around 180 °C. In addition, another peak at 500 °C was the aggregation of copper nanoparticles that gradually increased and dented to reform micro-particle size and grain growth. According to Figure 1b, the DTA analysis of micro-copper clay showed an exothermic peak at 340 °C, which is the sintering temperature of micro-copper particles. We could conclude that the synthesized nano- and micro-copper clay encourages the decrease in sintering temperature to lower than that of bulk copper metal [7]. Moreover, the DTA of both clays showed a strong peak when the temperature was increased to 1000 °C. Therefore, it can be implied that aggregated particles of copper suddenly restructured and became a solid form because the temperature was close to the melting point of copper.

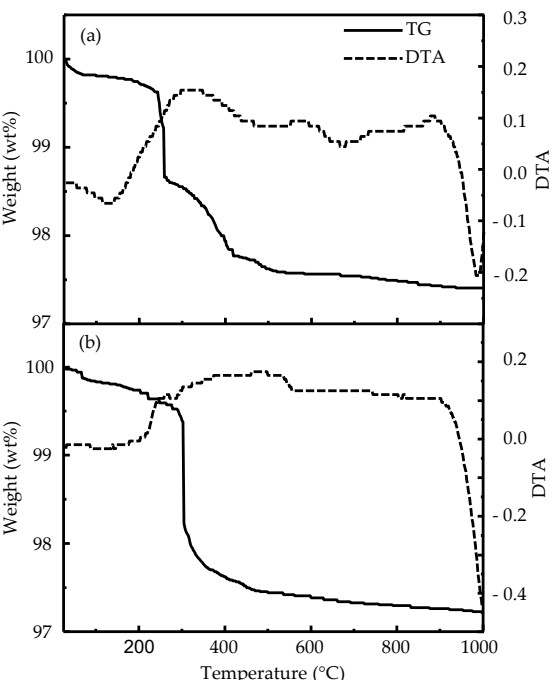

**Figure 1.** The TG and DTA curves of nano- (**a**) and micro-copper clays (**b**) fired at different temperatures.

### 3.2. Effect of Heating Temperature

To investigate the influence of heating temperature on the physical properties of the nano- and micro-copper clays, both clays were heated under varying heating temperatures from 200 to 1000 °C, whereas the same heating rate and holding time were 10 °C/min and 30 min, respectively. Figures 2 and 3 show the microstructure of the fired clays at different heating temperatures. For the fired nano-copper clay (Figure 2), the copper particles started sintering at the neck region when heated to 200 °C (Figure 2a). Afterward, the dumbbell shape of the interconnected nanoparticles was formed, and the sintering process could be assured (Figure 2b). When the nanoparticles were heated to 600 °C, the aggregated copper left many pores, seemingly corresponding to the schematic representation in Figure 2e. At this stage, it can be assumed that the copper atoms diffused from the nanoparticles at the intersections between the differently oriented particles continued to deposit onto the neck region [7,9]. Furthermore, as the temperature increased to 700 °C, the pores significantly decreased in size and became more isolated (Figure 2f). Thus, it is difficult to distinguish the particle shape from the neck regions, which enter sintering at the final stage [9,10]. Above 800 °C, the grain growth continued to form a dense, porous polycrystalline copper (Figure 2g). Finally, the copper atoms moved from the convex surface on one side of the grain boundary to the concave surface on the other side, resulting in a homogeneous phase of sintered materials. However, the sintered particle features collapsed and reformed after the fired temperature approached the melting point of bulk copper metal at 1000 °C (Figure 2i).

For micro-copper clay, the micro-particle was not sintered at 200 °C (Figure 3a). Comparing the micro- and nano-copper clay (Figures 2b and 3b), the micro-copper particles were connected to form a larger aggregated particle. The size of the pores significantly increased and became more isolated after the heating temperature reached >400 °C (Figure 3c). The fracture surfaces of the samples contained fewer dimples, yet many pores, including obturation due to fracture. The microcrack initiation tended to occur in areas containing pores, which rapidly propagated. After reaching the temperature of 700 °C (Figure 3f), the particle size and density correspondingly enlarged with a higher temperature; however, it also decreased the number of pores [10,11]. The fracture surface of the sintered samples showed the characteristics of the mixed fracture. Aside from the dimple fractures, intercrystalline

and transcrystalline cleavage fractures were also presented. Those features have resulted in increasing strength [12,13]. Comparing the results from Figure 2g,h and Figure 3g,h, the fracture surface of the sintered micro-copper clay indicated fewer pores and homogeneous tissue than nano-clay. It can be concluded that the particle size influenced the sintering densification process. The clay's microstructure became denser when the heating temperature was increased. Similar to nano-copper clay, the sintered particles of micro-copper clay collapsed and reformed, accompanied by several pores when the heating temperature approached 1000 °C (Figure 3i). Regarding heat temperature optimization, we concluded that micro-copper clay provided more sturdily and uniform densification than nano-copper clay compared to the same heating conditions. As a result, it could be affected by their physical properties, especially density and hardness.

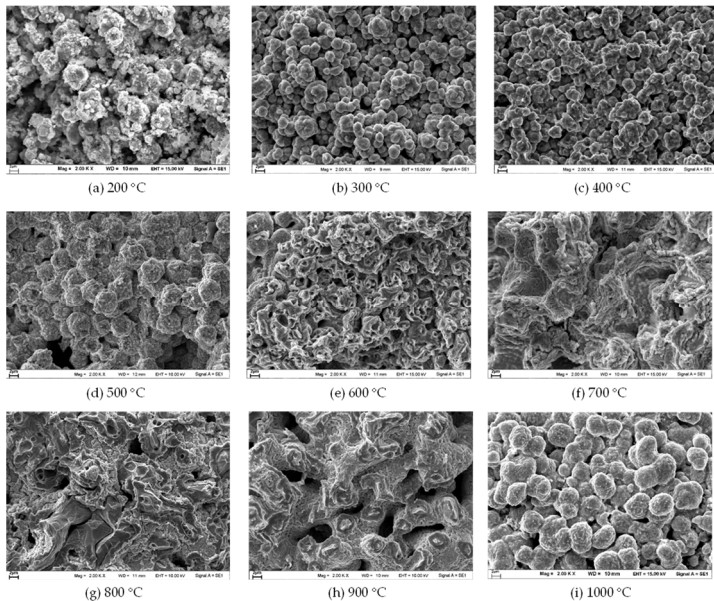

**Figure 2.** The SEM photographs of nano-copper clay fired at different temperatures: (**a**) 200 °C, (**b**) 300 °C, (**c**) 400 °C, (**d**) 500 °C, (**e**) 600 °C, (**f**) 700 °C, (**g**) 800 °C (**h**) 900 °C, and (**i**) 1000 °C.

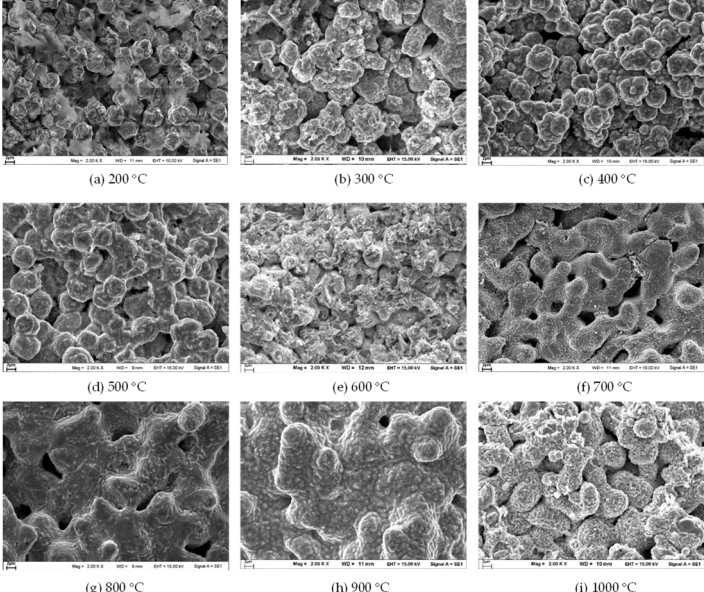

**Figure 3.** SEM photographs of micro-copper clay fired at different temperatures: (**a**) 200 °C, (**b**) 300 °C, (**c**) 400 °C, (**d**) 500 °C, (**e**) 600 °C, (**f**) 700 °C, (**g**) 800 °C, (**h**) 900 °C, and (**i**) 1000 °C.

Figure 4 shows the relationship between the volume shrinkage, bulk density, and hardness of the fired clays at different heating temperatures. Considering volume shrinkages at different heating temperatures (Figure 4a), dimensional changes occurred, resulting in volume shrinkage upon the complete removal of organic binders at about 200–600 °C. The volume shrinkage of the nano-copper clay was higher than the micro-copper clay at the same heating temperature because volume shrinkage alterations generally relate to the packing density of the metal particle during the densification process [13–15]. The results correspond well with the SEM images of nano- and micro-copper clays in Figures 2 and 3. Due to the high porosity between nanoparticles, the exceedingly small nano-copper particles drastically shirked to connect and provide higher volume shrinkage when the heating was performed. Therefore, the nano-copper clay has higher volume shrinkage than the micro-copper clay.

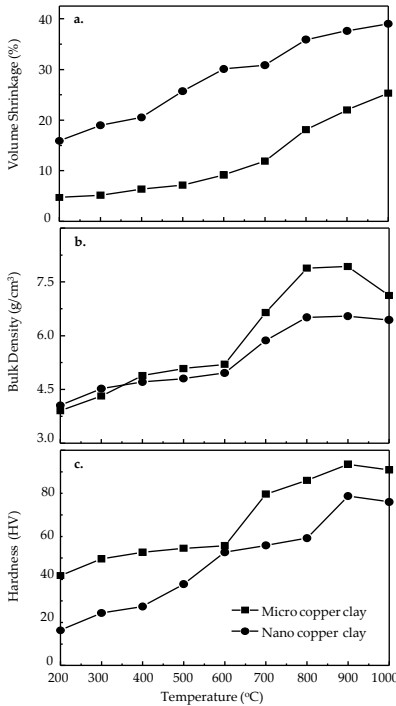

**Figure 4.** (**a**) Volume shrinkage, (**b**) bulk density, and (**c**) hardness of sintered nano- and micro-copper clays with different heating temperatures.

As mentioned previously, volume shrinkage is related to material density. When considering the density of the clays (Figure 4b), the figures show the same increasing trend of bulk densities by increasing heating temperatures. At the initial stage of 200–600 °C, both clays display a slight change in their bulk density due to the decomposition of water and organic binders. Upon the complete removal of the binders, Figure 4 shows a dramatic change in density when the temperature reaches 600–900 °C. The bulk density increased, corresponding to the decreasing porosity of the fired clay due to sintering densification [16,17]. In the 600–900 °C stage, micro-copper clay displays a higher bulk density than nano-copper clay because the packing density of the micro-copper clay during the densification process is greater than that of the nano-copper clay. SEM images at high firing temperatures verify these results in Figures 2 and 3. Moreover, the density of the micro-copper clay (about 8 g/cm$^3$) is close to the density value of bulk copper material, 8.96 g/cm$^3$. At 1000 °C, the density of nano- and micro-copper clays decreased, corresponding to their SEM images in Figures 2i and 3i, due to the sintered particles of both clays collapsing and deforming, accompanied by several pores when the heating temperature approached the melting point of copper metal (1085 °C).

Hardness is an important parameter in fabricating jewelry. As shown in Figure 4c, the hardness of both fired clays increased according to the rising heating temperature. The dramatic hardness change was observed at 600–900 °C when the copper particles were sintered and approached a complete densification process after losing organic binders and water. Since the hardness is related to the packing density of copper particles in its densification process [18,19], the results corresponded to the trend of bulk density at high fired temperatures (Figure 4b), as supported by SEM micrographs. The grain increased its size when the firing temperature increased. The fracture surface of sintered nano-clay exhibits open pores, leading to the relatively low porosity of the specimen at >800 °C. Consequently, the density and hardness of the nano-silver clay increased after 600 °C and was the highest at 900 °C.

It was remarkable that the hardness of the fired micro-copper clay was higher than that of nano-copper clay. However, the nanomaterial should have a higher hardness value than the micro-copper material [20]. It was suggested that thermally treating nanophase samples in the as-produced condition might result in structural changes, such as densification, as seen in the hardness of microstructures via SEM images. Moreover, some of the literature suggests that it may be due to artifacts from the sample preparation or the heating conditions performed [5]. Since copper particles are sensitive to oxidization [21], the chemical composition of copper clay fractures was also determined by SEM–EDS to clarify some artifacts in specimens. In Figure 5, the percentage by weight of sulfur (S) and oxygen (O) in nano-copper clay is higher than micro-clay in every procedure performed. Consequently, oxidation might discourage the deformation of particles in the densification process [7] and affect the hardness of copper clay. Therefore, micro-copper clay has better physical properties than nano-copper clay under the same firing temperatures.

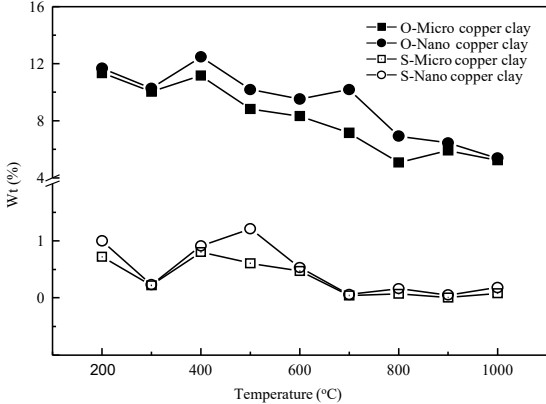

**Figure 5.** The percentage by weight of sulfur (S) and oxygen (O) in the sintered nano- and micro-copper clay specimens determined by SEM-EDS.

### 3.3. Effect of Heating Rate

To investigate the effects of heating rate, different heating rates, including 10 °C/min, 30 °C/min, and 60 °C/min, were performed at the same heating temperatures and holding times. The SEM images of nano-copper clay in the micro-features of fired specimens changed with the varying heating rates, as shown in Figure 6. When the heating rate increased from 10 to 30 °C/min, the microstructure decreased the densification with a few sintering necks. With another increase in the heating rate, up to 60 °C/min, the nano-copper particle sintered, and its effects became more pronounced on porosity and densification, indicating that the high heating rate could not provide sufficient time for improving the packing density and densification of the microstructure [13,17].

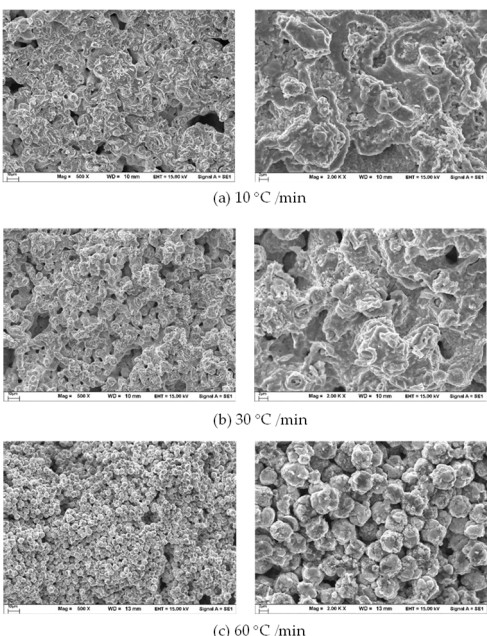

**Figure 6.** SEM images of sintered nano-copper clay with different heating rates at (**a**) 10 °C/min, (**b**) 30 °C/min, and (**c**) 60 °C/min.

The SEM images of sintered micro-copper clay at different heating rates are shown in Figure 7. The microstructure shows the sintered particles and grain growth where densification was more pronounced at the lowest heating rate (10 °C /min). On the other hand, the results showed that a high heating rate of 60 °C/min could not use the uniformization of grain growth because the process is quick to employ any densification. Therefore, both sintered nano- and micro-copper clays firing under a high heating rate have higher porosity and poor particle packing than low heating rates.

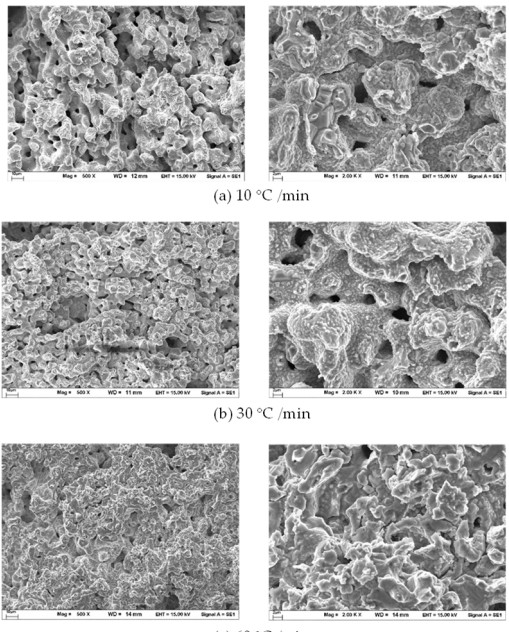

**Figure 7.** SEM images of sintered micro-copper clay with different heating rates at (**a**) 10 °C/min, (**b**) 30 °C/min, and (**c**) 60 °C/min.

Comparing the SEM images of nano- and micro-copper clay at 10 °C/min (Figures 6a and 7a) and 30 °C/min (Figures 6b and 7b), there are a few different grain sizes in the sintered

sample performing at these heating rates. On the other hand, the SEM images at the 60 °C/min heating rate showed dramatically different grain sizes (Figures 6c and 7c). As for the results, it can be confirmed that the heating rate has a profound effect on the final grain grown compared to the starting particle size of both nano- and micro-copper clays. Thus, complete densification can be achieved with a low heating rate.

Figure 8a shows the volume shrinkage versus the heating rate operated on the sintered samples. Notably, the shrinkage rate depends on the copper clay's heating rate and particle size as they increase, resulting in decreased shrinkage [9,22,23]. Moreover, the lowest heating rate (10 °C/min) produced a higher shrinkage because the sample remained in the furnace and reached its isotherm temperature for a long time. The maximum shrinkage rate was observed at a low heating rate of 10 °C/min and low shrinkage with the increased heating rate. The samples sintered with heating rates of 10 and 30 °C/min yielded a relatively high density and hardness value compared with 60 °C/min, as shown in Figure 8b,c.

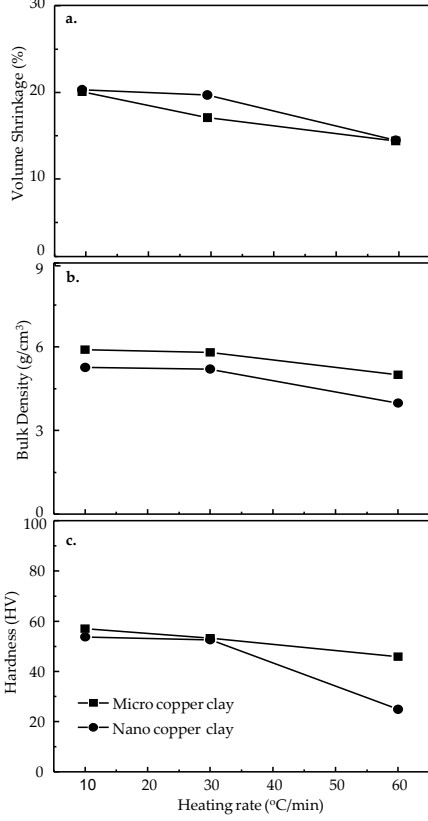

**Figure 8.** (**a**) Volume shrinkage, (**b**) bulk density, and (**c**) hardness of sintered nano- and micro-copper clays with different heating rates.

By comparing the bulk density of sintered nano- and micro-copper clays fired at different heating rates (Figure 8b), slow heating rates induce a high density. The bulk density of both clays fired at 10 °C/min was slightly better than 30 and 60 °C/min. Our results are supported by SEM micrographs of the fired nano- (Figure 6) and micro-copper clays (Figure 7). When the heating rate increased at a constant temperature, the porosity increased with the increase in the heating rate due to the sintering densification process [24,25]. Considering the heating rate at 60 °C/min (Figures 6c and 7c), the fracture surface of the sintered nano-clay exhibited open pores, leading to the specimen's relatively low density.

Figure 8c shows the hardness of the fired nano- and micro-copper clays, which decreases as the heating rate increases. The results correspond with the SEM images at high firing temperatures (Figures 6 and 7). However, the pore size affects the bulk density and microhardness of the material. The largest pore was observed when the heating rate was

60 °C/min, correlating with the lowest hardness (Figure 8). Consequently, the density and hardness of micro-copper clay were not significantly different. Nevertheless, the change in heating rate highly influences the size of nanoparticles than microparticles. In the case of nano-copper clay, the hardness is not affected after the heating rate changes from 10 to 30 °C/min. However, a dramatic change in hardness was observed when the heating rate reached 60 °C/min, corresponding to SEM micrographs (Figure 6c). Since hardness is an important factor in jewelry products, we could conclude that the optimal heating rate for the nano- and micro-copper clays was 10 °C/min.

### 3.4. Effect of Holding Time

The effect of holding time was investigated by controlling the same heating temperatures and heating rates at 800 and 10 °C/min, respectively. The holding times were varied for 0, 30, and 60 min. As illustrated in Figures 9 and 10, the extent of the holding time tends to improve the densification of both clays. In the case of the holding time at 0 min (Figures 9a and 10a), the micro- and nanoparticles were sintered and grew to a bigger micron size; however, complete densification did not occur. When the holding time was set to 60 min (Figures 9c and 10c), the SEM images of both nano- and micro-copper clays revealed grain growth in a uniformed micron structure with an even surface. Moreover, the obtained grain growth of micro-copper clay is smoother than nano-copper clay.

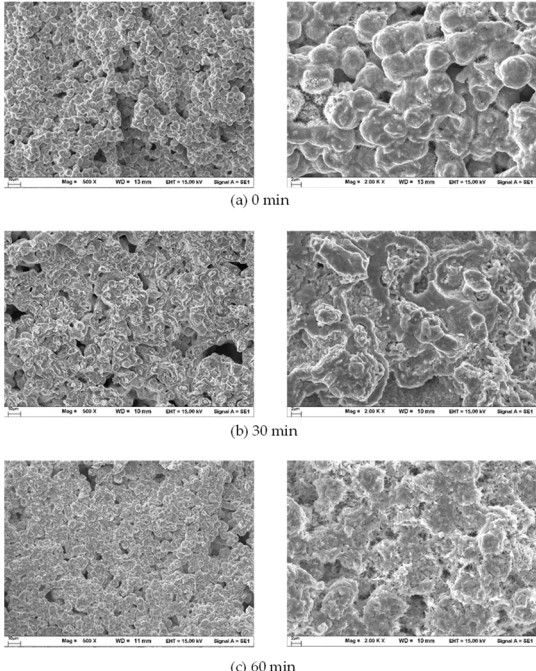

**Figure 9.** SEM images of sintered nano-copper clay with different heating times at (**a**) 0 min, (**b**) 30 min, and (**c**) 60 min.

Figure 11a shows the extent of holding times, which increase the volume shrinkage in both nano- and micro-copper clays. At 0 min of holding time, nano- and micro-copper clays revealed similar volume shrinkage at about 5%. At 30 and 60 min of holding time, the volume shrinkage of the nano-copper clay was higher than micro-copper clay. These results may be due to the influence of particle size on the volume change. The decomposition of the binders and densification phenomena led to an increase in the volume shrinkage percentage. A long holding time is unnecessary for the sintering process in jewelry fabrication because of high volume shrinkage [4], especially in the case of nano-copper clays. However, the hardness of the specimens must be considered in this application.

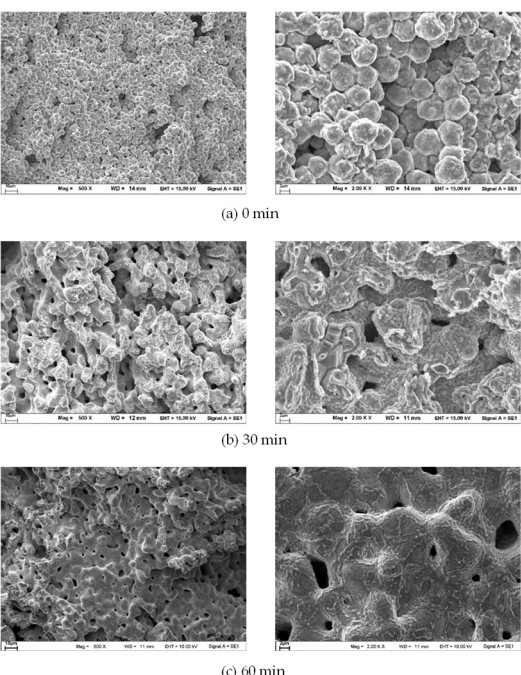

**Figure 10.** SEM images of sintered micro-copper clay with different heating times at (**a**) 0 min, (**b**) 30 min, and (**c**) 60 min.

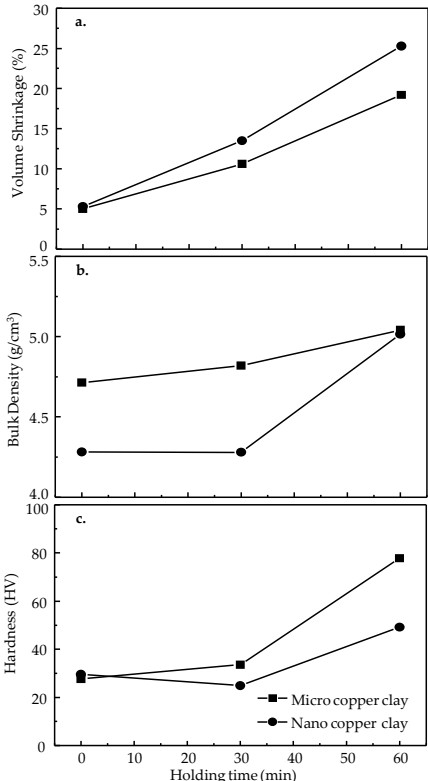

**Figure 11.** (**a**) Volume shrinkage, (**b**) bulk density, and (**c**) hardness of sintered nano- and micro copper clay with different holding times.

Figure 11b reveals the bulk density of nano- and micro-copper clays fired at different holding times. The extent of holding time tends to improve the density of the finished product of nano- and micro-copper clays. At the holding time of 0 min, the nano-copper clays revealed very low bulk density compared to the micro-copper clays. After the

holding time increased to 30 min, the micro-copper clay showed an increase in density. The nano-copper clay revealed a slight decrease in the bulk density, likely due to the copper nanoparticles becoming necked and sintered after prolonging the holding time (Figure 9). Since the number of open surface pores reflected the decline in density [25–27], the density of the nano-copper clays was lower than that of the micro-clay when the holding time was extended to 30 min. After the holding time approached 60 min, the sintered copper microparticles of micro- and nano-copper clays were connected and obtained grain growth with a uniform micron structure and even surface. Although the bulk densities of both clays were almost equal at longer holding times, a longer holding time is necessary for the sintering process of jewelry fabrication by copper clay, especially for nano-copper clays. Therefore, in the case of jewelry fabrication, considering only volume shrinkage, density, and hardness of the final product is not enough.

Considering the relation between holding time and hardness of nano- and micro-copper clays in Figure 11c, the hardness trend is similar to those of bulk density in Figure 11b. However, micro-copper clay has a higher hardness than nano-copper clay. Herein, it is well known that hardness relates to the packing density of particles in their densification process [4,8]. Although the density of nano-copper clay is similar to micro-clay with an extended holding time of 60 min, the hardness of the micro-copper clay is higher than nano-copper clay at the same holding time of the firing process. The results corresponded to their SEM images (Figures 2 and 3), in which the small nano-clay particles are more porous than micro-clay. The EDS results in Figure 5 also pointed out that the nano-copper clay shows a high oxidation element. Nevertheless, a longer holding time is necessary to fire both copper clays, especially in jewelry applications. Although the longer the holding time revealed high shrinkage, it exposed better mechanical properties for jewelry wearing, such as hardness and density.

### 3.5. The Nano- and Micro-Copper Clay for Jewelry Fabrication

We produced nano- and micro-copper clays from synthesized nano- and micro-copper particles and used them to fabricate jewelry prototypes (Figure 12). The optimal firing conditions were selected from the previous microstructure and physical property measurements: a temperature >800 °C, a heating rate of 10 °C/min, and a holding time of 60 min. However, the final products showed that nano-copper clay jewelry offers a higher shrinkage volume than micro-copper clay (Figure 12). Although nano-copper clay provides a lower density and hardness than micro-copper clay, it can be used to create wearable jewelry under these heating conditions. Furthermore, nano-copper clay applies to other industrial applications, such as copper paste, copper ink, and ceramic slip, because it has a low sintering temperature and good physical properties.

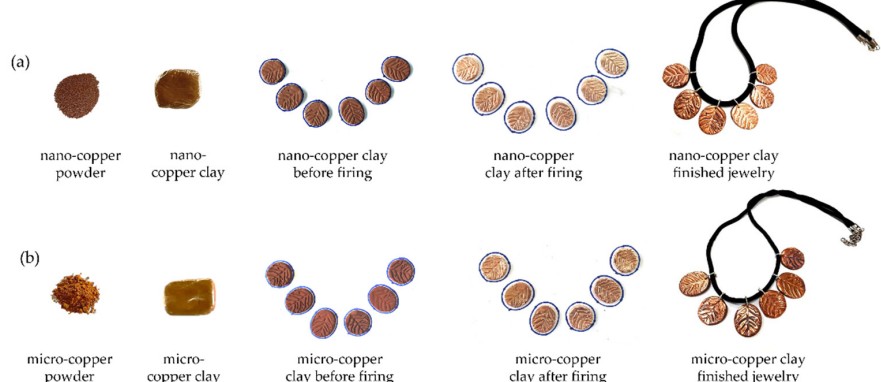

**Figure 12.** (**a**) Synthesized nano-copper particles (left) made into finished nano-copper clay jewelry (right); (**b**) Synthesized micro-copper particles (left) made into finished micro-copper clay jewelry (right). The blue line shows the initial size of each specimen before the firing process.

## 4. Conclusions

Nano- and micro-copper particles were synthesized by chemical reduction to prepare nano- and micro-copper clay. The clays were produced by mixing the same weight of copper particles, organic binders, and water. Using the proper ratio, both nano- and micro-copper clays can be formed for molding, shaping, pressing, or sculpting for jewelry or artwork. The sintering process of nano-copper clay was initiated at 180 °C, lower for micro-copper clay. Although nano-copper particles can be sintered at low temperatures, high firing temperatures provide good mechanical properties for wearable jewelry. Under the same heating conditions, the micro-copper clay had better mechanical properties than nano-copper clay because an artifact of copper oxidation in the nano-copper clay affected the particle densification process. Moreover, the volume shrinkage of nano-copper clay is greater than micro-sized copper clay at the same heating conditions due to nano-copper clay's small particle size. Volume shrinkage is an important factor when producing handmade jewelry from metal powder. The finished product will shrink after firing because of metal particle sintering and weight loss of water and organic binders. As a result, molded jewelry must be sized up to ensure that the finished piece is the intended size, especially for rings or bracelets. Regarding physical properties, including hardness, shrinkage, density, and the porosity of wearable jewelry products, the optimal heating conditions for both nano- and micro-copper clays should be >800 °C, with a heating rate of 10 °C/ min for 60 min. Nano-sized copper induces high shrinkage of the fired specimen, but jewelry products use nano-sized particles to achieve fine textural details and high hardness after firing under optimal conditions. However, heating the samples under a vacuum or inert gas atmosphere to minimize oxidization is recommended.

**Author Contributions:** Conceptualization, P.T. and M.J.; methodology, P.T., D.M., and W.R.; validation, P.T. and M.J. formal analysis, P.T.; investigation, P.T.; resources, P.T.; data curation, P.T.; writing—original draft preparation, P.T.; writing—review and editing, P.T., M.J., and A.P.; visualization, P.T.; supervision, P.T.; project administration, P.T.; funding acquisition, P.T. All authors have read and agreed to the published version of the manuscript.

**Funding:** This research was funded by the National Research Council of Thailand (NRCT), grant number 31/2555.

**Institutional Review Board Statement:** Not applicable.

**Informed Consent Statement:** Not applicable.

**Data Availability Statement:** All the data are presented in the main text. Any other data are available from the corresponding author upon reasonable request.

**Acknowledgments:** The authors gratefully acknowledge the National Metal and Materials Technology Center Thailand (MTEC), the Microscopic Center, Faculty of Science, and Faculty of Engineering, Burapha University for their instrumental support in STA analysis, SEM analysis, and micro-hardness measurement, respectively.

**Conflicts of Interest:** The authors declare no conflict of interest.

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
