# Peer review of "Effect of Thermal Treatment on Nano- and Micro-Copper Particles for Jewelry Making"

_applsci, doi:10.3390/app122312050_

Round 1

Reviewer 1 Report

In this paper, the nano- and micro-sized copper particles were synthesized and mixing with the same organic binders and water content to produce an alternative clay-like material for making handmade jewelry and artworks. The authors addresse the effect of thermal treatment on thermal behavior and the development of physical properties of different size copper particles. Overall, the article is well organized and its presentation is good. However, some minor issues still need to be improved:

(1) The authors should provide the synthesis and characterization methods of copper particles with different sizes, since they mentioned the specific size in the paper. “The average size of those nanoparticles can be controlled in the range of 120-200 nm, while the micro-copper particles are about 3-5 micron.”

(2) The specific composition and fabrication methods of nano- and micron-copper clays should be disclosed in detail so that they can be replicated by those in the same field.

(3) It is desirable to provide some photos of the synthesized copper particles, nano- and micro-copper clay before and after sintering, so that the reader can visualize the whole fabrication process.

(4) Compared with silver, copper is very easy to get oxidized, especially in the nanoscale, so it is recommended to add experimental results of sintering under vacuum or inert gas atmosphere.

Reviewer 2 Report

The authors studied the effect of heating on the physical properties of the nano- and micro-sized copper particles mixed with organic binders and water, thus producing a clay-like material.

1. The title of the article does not correspond to its contents because there is no reference to the clays, whereas the clays are the materials studied in this work.

2. The level of English used in the article is extremely low, there are grave spelling, grammar, vocabulary mistakes virtually in every sentence.

3. The performed literature review is inadequate.

4. It was not explained why the conclusions on the higher shrinkage of clay based on the copper nanoparticles with respect to the clay based on microparticles are important for jewelry fabrication. 

5. In the conclusion, the authors write "jewelry product takes advantages of the nano-sized of copper to give a fine detail of textured and good appearance". What is a fine detail of textures? What is a good appearance? How can anybody differ a good appearance from a bad one? These are the conclusions pertinent to the field of art, it is not science, as these conclusions are not and cannot be corroborated by any quantitative measurement.

6. It is recommended to the authors to continue their research of clays based on nano- and microparticles of metals. It would be an interesting idea to study these clays synthesized with various organic linkers and with various metals. Nevertheless, it is advised to perform a rigorous study of existing literature and to find an adequate knowledge gap that can be addressed in their future study.

Round 2

Reviewer 2 Report

Publish as it is.

Author Response

Your Sincerely,

P.Thongnopkun
